# The Importance of Managing Modifiable Comorbidities in People with Multiple Sclerosis: A Narrative Review

**DOI:** 10.3390/jpm13111524

**Published:** 2023-10-24

**Authors:** Viviana Nociti, Marina Romozzi

**Affiliations:** 1Centro Sclerosi Multipla, Fondazione Policlinico Universitario Agostino Gemelli IRCCS, 00168 Rome, Italy; marina.romozzi@jefferson.edu; 2Dipartimento Universitario di Neuroscienze, Università Cattolica del Sacro Cuore, 00168 Rome, Italy

**Keywords:** MS, comorbidities, personalized medicine, disease-modifying therapies

## Abstract

Multiple sclerosis (MS) is a chronic, inflammatory, degenerative demyelinating disease of the central nervous system (CNS) of unknown etiology that affects individuals in their early adulthood. In the last decade, life expectancy for people with MS (PwMS) has almost equaled that of the general population. This demographic shift necessitates a heightened awareness of comorbidities, especially the ones that can be prevented and modified, that can significantly impact disease progression and management. Vascular comorbidities are of particular interest as they are mostly modifiable health states, along with voluntary behaviors, such as smoking and alcohol consumption, commonly observed among individuals with MS. Vascular risk factors have also been implicated in the etiology of cerebral small vessel disease. Furthermore, differentiating between vascular and MS lesion load poses a significant challenge due to overlapping clinical and radiological features. This review describes the current evidence regarding the range of preventable and modifiable comorbidities and risk factors and their implications for PwMS.

## 1. Introduction

Multiple sclerosis (MS) is a chronic, inflammatory and degenerative demyelinating disease of the central nervous system (CNS) of unknown etiology [1]. It typically affects individuals in early adulthood. The disease displays significant variability in its clinical manifestations, subtypes, and outcomes [1]. Clinically isolated syndrome (CIS) describes a first episode of neurological symptoms consistent with a demyelinating etiology and suggestive of MS. Nearly 85% of patients present with a relapsing-remitting MS (RRMS) disease type at onset, defined by episodes of acute exacerbations followed by recovery. Most RRMS patients convert to secondary progressive MS (SPMS), characterized by a gradual decline in neurological functions with or without relapses. In 5–10% of cases, patients present a primarily progressive course (PPMS), where disability progressively worsens from the beginning of the disease [1].

The pathophysiology of MS involves abnormal immune activation, with various immune cells in the periphery and resident CNS cells contributing to neuroinflammation. However, the precise trigger for this immune response remains unknown [2,3]. It is plausible that the neuroinflammatory component remains a major driver of MS pathogenesis, causing myelin sheath destruction and, ultimately, axonal damage [4]. However, a significant amount of evidence has shown that inflammation is not entirely detrimental [5]. Previous studies have documented the reparative activities of the inflammatory response in MS. In these circumstances, immune cells release neurotrophic factors and engage with neurons and glial cells to protect them from damage, promoting growth and repair [6].

The treatment landscape of MS has changed significantly in the past decades due to the growing number of approved disease-modifying therapies (DMTs), which can influence the disease course by preventing relapses and disability progression in people with MS (PwMS) [7]. Furthermore, with a better understanding of the pathophysiological pathways that drive progressive disease, immunomodulatory therapies that simultaneously mitigate acute and chronic neuroinflammation are actively being studied in later-phase clinical trials for both relapsing and progressive MS. Additionally, various approaches aiming to promote remyelination have also been explored [8].

In recent decades, the MS population has been aging in parallel with increasing general life expectancy, advances in DMTs, and improved health and social care, and MS has become a lifelong condition. The risk of multi-morbidity increases with age; however, the accrual of comorbidities over time in these patients could be explained by several other factors, such as physical invalidity, a background of common risk factors, and the use of DMTs [3]. Comorbidity refers to the co-occurrence of multiple diseases or medical conditions in a patient, which may have similar underlying causes and impact the course of the diseases themselves [9].

Multisystem comorbidities are common in PwMS, predominantly neurological disturbance, psychiatric comorbidities, vascular comorbidities (e.g., hypertension, hyperlipidemia), chronic lung diseases, metabolic disorders and autoimmune conditions [10,11,12].

Cardiovascular comorbidities, such as abnormalities in blood pressure, heart rhythm, and left ventricular systolic function, are frequent in PwMS. Having MS also heightens the risk of myocardial infarction, stroke, and heart failure [13,14]. 

A study involving 8983 PwMS enrolled in the North American Research Committee on Multiple Sclerosis (NARCOMS) Registry revealed that vascular comorbidities, whether present at symptom onset, diagnosis, or later in the disease course, were associated with an elevated risk of disability progression in PwMS [15]. Having just one vascular comorbidity at the time of diagnosis was linked to a 51% increased risk of early gait disability, while having two such comorbidities was associated with a 228% increased risk [15].

Many comorbidities manifest before or at MS symptom onset, but their prevalence tends to rise over time [16]. For instance, some comorbidities, such as depression and anxiety, may represent preclinical symptoms of MS arising from immunological and inflammatory alterations in the CNS [11]. 

Multiple studies have shown that the presence of comorbidities can determine a delayed diagnosis, greater severity of the disability at MS diagnosis, and an increased relapse rate [17]. Furthermore, comorbidities can affect treatment choices, quality of life, and mortality [18,19,20]. 

Comorbid conditions exacerbate the disability burden experienced by PwMS, increase the complexity of patients’ management, and have important clinical and socioeconomic implications [18]. A significant proportion of comorbidities can be prevented and modified, providing important opportunities for intervention and reduction of the disease burden (Figure 1) [21].

This review aims to describe the current evidence regarding the range of preventable and modifiable comorbidities and risk factors and their implications for PwMS.

## 2. Methods

A comprehensive literature search was conducted across multiple databases to identify relevant studies on MS and modifiable risk factors/comorbidities, published up to June 2023. Furthermore, we manually checked the references of relevant papers and reviews to find any other pertinent articles. Finally, the extracted data were organized into spreadsheets.

## 3. Hypertension

Hypertension is one of the most frequent comorbidities among PwMS, with a reported prevalence of up to 47% [22]. While the prevalence of hypertension at the time of MS diagnosis ranges from 0 to 8% [22].

A cohort study involving 8983 PwMS enrolled in the NARCOMS Registry reported that 30.1% of MS participants had hypertension, similar to the rate expected for the general population [15]. A cross-sectional study based on electronic health records evaluated the prevalence of hypertension in a large sample of 122,660 PwMS and 37,075,350 non-MS individuals [23]. After adjusting for age and sex, the study found that hypertension was significantly more prevalent in the MS group compared to the non-MS group [23]. Specifically, hypertension was 25% more common among individuals with MS, and its prevalence increased with age. Furthermore, the study highlighted that hypertension was more prevalent in black Americans and males [23]. Additional data suggests that ethnicity plays a role in the prevalence of hypertension among PwMS, with rates of 28.9% in black Americans and 16.5% in white Americans [24]. 

Edwards et al. conducted a study to evaluate the prevalence of comorbidities in PwMS from 2006 to 2014. Their findings revealed that hypertension was the second most common comorbidity, affecting over 25% of patients within an individual year. The percentage of patients with hypertension generally increased from 2006 to 2013 but declined from 2013 to 2014 [25]. Another study examining cardiovascular conditions in 1548 individuals with MS found that they had a 48% higher likelihood of having a history of hypertension compared to controls [26].

A retrospective study on 2926 PwMS found no significant difference in the prevalence of hypertension between individuals with and without MS, overall or when stratified by sex. One of the study’s main findings was that MS was not associated with worse blood pressure control, and the mean number of medications used to treat hypertension did not differ between individuals with and without MS [27]. 

Regarding sex differences, a Canadian study involving 23,382 MS cases reported that the prevalence of hypertension at the time of MS diagnosis was 16% higher in women with MS and 48% higher in men with MS compared to matched healthy controls [28]. Another study focused on male veterans with MS and found a hypertension prevalence of 47% [29]. 

Stenberg et al., in a study comparing 188 PwMS taking interferon-beta (IFN-β), glatiramer acetate or natalizumab with 110 PwMS naïve to drugs, found that the use of DMTs has been associated with higher diastolic blood pressure readings [30].

However, contrasting results were reported in other studies. A study on hospitalized PwMs found that MS patients were significantly less likely to have risk factors, including hypertension, hypercholesterolemia, diabetes and obesity [31]. The same findings were reported in another study on hospitalized elderly MS patients [32].

Some studies investigated the relationship between hypertension and disability in PwMS. In a 3-year longitudinal study on 4358 patients with RRMS, hypertension emerged as the most significant factor that negatively impacted the clinical outcomes and the walking speed [33].

A study by Marrie et al. showed that hypertension was one of the comorbidities independently associated with an increased risk of ambulatory disability regardless of the specific endpoint chosen [15].

To summarize, the prevalence of hypertension is high in the general population, and it ranks among the leading risk factors for disability [34]. Therefore, establishing whether hypertension is more prevalent in PwMS compared to the general population is challenging. However, emerging evidence suggests that hypertension as a comorbidity in MS patients may be linked to a more unfavorable progression of disability [15,33]. It is plausible that the higher prevalence of hypertension in this population can be attributed to several factors, including chronic inflammation, the use of corticosteroids in managing MS exacerbations, and potentially the use DMTs [30,35]. In addition, lifestyle factors, such as reduced physical activity and weight gain due to decreased mobility, may also contribute to the development of hypertension in PwMS.

## 4. Type-2 Diabetes

The prevalence of type-2 diabetes in PwMS ranges from 6.7% to 8.5% [22]. Studies have reported diabetes prevalence at or before the diagnosis of MS ranging from 0.8% to 5.6% [22]. Data from the NARCOMS Registry indicated a prevalence of diabetes of 6.1% [15].

The results from various studies comparing the prevalence of diabetes in PwMS to a control population have been inconsistent, indicating that the prevalence of diabetes can be higher, lower, or similar in the MS population [28,32,36,37].

Marrie et al. found that the age-adjusted prevalence of type-2 diabetes in PwMS (7.6%) was similar to that of the general population (8.31%) [36]. In another study, the prevalence of type-2 diabetes in a cohort of 1206 MS patients was significantly higher than the general population (6.75%). In this group, 35% of individuals experienced diabetes prior to being diagnosed with MS. Conversely, among those who developed diabetes after being diagnosed with MS, 41.5% received their diabetes diagnosis within the initial 5 years [37].

Maric et al. found that type 2 diabetes was significantly more prevalent in patients with PPMS than patients with RRMS. The authors suggested this difference could be attributed to the slightly higher age of PPMS patients [38]. On the contrary, Fleming and Blake observed a lower prevalence of type-2 diabetes in a cohort of hospitalized elderly MS patients than in controls [32].

Hou et al. carried out a 9-year population-based study to examine the incidence and relative risks of MS in individuals with type-2 diabetes. Their findings revealed a significant association between type-2 diabetes and a heightened risk of developing MS, with a hazard ratio of 1.44 [39]. 

Marrie et al. conducted a retrospective study on the adequacy of disease control and treatment intensity for diabetes on 2926 MS people, 298 (10.2%) of those with diabetes. The mean number of medications used for diabetes was not different between individuals with MS and without MS. They concluded that MS did not adversely affect the control of diabetes [27].

Some studies assessed the interplay between disability and diabetes in MS patients. Data from the NARCOMS Registry demonstrated that at any time of MS course, diabetes was associated with an increased risk for requiring unilateral and bilateral assistance for ambulation (HR 1.28 and HR 1.56, respectively) [15]. Conway et al. studied the impact of several comorbidities on disease courses in RRMS patients. PwMS with diabetes (5.8% of the cohort) had walking speeds that were 0.15 ft/s slower than MS patients without diabetes [33]. 

Diabetes has also been shown to impact cognitive functioning in PwMS. In a study by Marrie et al., in a cohort of 111 PwMS, diabetes was associated with reduced visual memory and verbal fluency [40]. Diabetes could also impact MS disability in an indirect way causing decreased sensation, neuropathic pain, and reduced vision [15,21].

## 5. Obesity

Several studies demonstrated that the prevalence of overweight and obesity in the MS population exceeds 50% [41,42]. In a study by Pilutti et al., the prevalence of these conditions almost reached 70% [43]. In a survey on 8983 PwMS from the NARCOMS registry, Marrie et al. found that a quarter of participants were obese and 31% were overweight [44]. However, the prevalence of obesity has been shown not to be significantly different from non-diseased adult people [41,45]. 

The association between body mass index (BMI) and the risk of developing MS has been explored in multiple studies, finding an association between obesity during childhood and adolescence and the risk of developing MS [46,47,48,49]. However, no significant associations have been found between BMI during adulthood and MS risk [49]. 

Several studies assessed the reciprocal association between obesity and disability in PwMS and the impact of obesity on disease progression with conflicting results. In a study involving women with MS, patients with less mobility had significantly greater total and abdominal fat accumulation [41]. Tettey et al. found that higher BMI was independently associated with a higher Extended Disability Status Scale (EDSS) [50].

Data from the NARCOMS registry showed that, after accounting for smoking status and physical activity level, MS participants with severe disability had decreased odds of being overweight and obese [44]. The proportion of underweight participants (BMI < 18) increased with worsening disability status [44]. Pilutti et al. examined the effect of weight status on mobility in 168 ambulatory PwMS, finding no significant impact of BMI on mobility outcomes [43]. In another study on the associations between BMI and relapse in 141 RRMS participants followed for 3 years, BMI was not associated with the hazard of relapse [51]. 

Fitzgerald et al. evaluated visceral adiposity by measuring waist circumference in 5832 PwMs. Of these, 3181 (55%) reported waist circumference meeting the criteria for the abdominal obesity component of metabolic syndrome. In multivariable models adjusting for overall obesity status, waist circumference was associated with 47% increased odds of severe versus mild disability [52]. 

Obesity is characterized by a low-grade chronic inflammation in which proinflammatory mediators, known as adipokines, are secreted from the adipose tissue [53]. The family of adipokines include several mediators, of which leptin and interleukin-6 have been shown to reduce regulatory T-cell activity and promote an inflammatory T helper 1 cell that mediates the immune-mediate inflammation in MS (Figure 2) [54,55].

In conclusion, obesity and overweight in childhood and adolescence increase susceptibility to MS. The correlation between weight status and disability has not yielded conclusive findings. A combination of factors, including reduced physical activity and impaired mobility can contribute to weight gain in MS patients. Moreover, the chronic inflammation associated with obesity and MS may lead to a synergistic effect, exacerbating neuroinflammation [53]. Obesity also contributes to cardiovascular comorbidities, such as hypertension and dyslipidemia, which are already prevalent in MS patients. 

## 6. Smoke

In a survey of 8983 PwMS enrolled in the NARCOMS registry, nearly 55% of participants reported ever smoking. One-third of participants (2747) smoked at the diagnosis of MS, and 1542 (17.3%) currently smoke [44].

Evidence from multiple studies showed that cigarette smoking is a risk factor for developing MS. A Canadian study detected an increase in MS risk in ever-smokers compared with never-smokers (odds ratio [OR] 1.6; 95% confidence interval (CI) 1.0–2.4). For heavy smokers (20–40 cigarettes per day), the risk increased almost 2-fold [56]. In a nested case–control study on PwMS patients, the proportion of ever-smokers before the index date was 45.8% among cases and 39.4% among controls [57]. When compared to non-smokers, the OR (95% CI) for MS was 1.3 (1.0–1.7) for ever-smokers, 1.4 (1.0–1.9) for current smokers, and 1.0 (0.6–1.8) for former smokers. This association was consistent across different clinical presentations of MS, including RRMS and PPMS [57].

In contrast, another study by Healy et al. revealed that individuals who were current smokers (adjusted OR 2.42; 95% CI 1.09–5.35) or ex-smokers (adjusted OR 1.91; 95% CI 1.02–3.58) had a higher likelihood of developing PPMS compared to an initially relapsing course (RRMS or SPMS at baseline visit) when compared to never smokers [58]. Additionally, current smokers exhibited more severe disease characteristics at baseline, as indicated by elevated scores in EDSS, Multiple Sclerosis Severity Score, and brain parenchymal fraction, a marker of overall brain atrophy [58]. 

A meta-analysis by Hawkes et al. of six studies published in 2007 demonstrated that smoking was a significant risk factor for the subsequent development of MS, with an overall OR of 1.25–1.51 [59]. Handel et al. updated the meta-analysis by Hawkes et al. [59]. Their analysis incorporated data from 14 studies involving 3052 MS patients and 457,619 controls. They employed a conservative approach (examining smoking behavior before MS onset) and a nonconservative approach (regardless of whether smoking occurred before or during MS onset). In both models, smoking was linked to an increased susceptibility to MS. However, these studies exhibited significant heterogeneity [60]. A systematic review and meta-analysis using the Bradford Hill criteria for causation found that smoking had a statistically significant association with both MS and SPMS risk [61].

The interplay between smoking and progression is currently very controversial, with some studies suggesting an increased risk of progression. A meta-analysis by Hempel et al. on seven studies found that smokers had an increased risk of MS progression (HR 1.55; CI 1.10–2.19) [62]. Hernán et al. demonstrated that in 179 cases with RRMS clinical onset, the HR of secondary progression was 3.6 (95% CI 1.3–9.9) for ever-smokers compared with never-smokers [57].

A prospective cohort study of people with MS in Southern Tasmania from 2002 to 2004 showed that cumulative cigarette pack years smoked after cohort entry was associated with an increase in longitudinal Multiple Sclerosis Severity Score [63].

In the longitudinal analysis of the study by Healy, it was observed that MS patients who smoked experienced a faster progression from RRMS to SPMS compared to non-smokers. In addition, in smokers, the T2-weighted lesion volume increased faster, and the brain parenchymal fraction decreased faster [58].

Briggs et al. found that patients who were active or former smokers exhibited more significant impairment in ambulation and global disability, along with higher depressive scores, in comparison to those who never smoked [24].

A study by Manouchehrinia et al. found that disease progression was more rapid amongst ever-smokers. The authors also assessed the effects of smoking cessation on disability progression in PwMS. Ex-smokers demonstrated a significantly reduced risk of reaching EDSS score milestones 4 and 6, irrespective of whether they quit smoking before or after MS onset [64]. Likewise, a retrospective study found that the adverse impact of tobacco smoke on EDSS lessened after smoking cessation, and the reduction in the risk of disability progression was time-dependent [65].

The link between smoking and MS risk might be attributed to the susceptibility of oligodendroglia to nitric oxide [66]. Moreover, serum levels of cyanide, a component of cigarette smoke, and its main metabolite, thiocyanate, have been known to cause demyelination in the CNS of animals for a long time [67]. Other proposed mechanisms connecting smoking and MS involve the pro-inflammatory effects of cigarette smoke components, the predisposition to immune-mediated responses in smokers, the direct impact of cigarette smoke components on the blood–brain barrier (BBB), and the increased frequency and persistence of infections due to smoking (Figure 2) [68,69]. In addition to the direct effects of MS-specific disease progression, the accumulation of disability can also occur indirectly through the effect of smoking on respiratory function, which can result in a decline in mobility and other activities of daily living [70].

To summarize, smoking habits are significantly associated with the development and progression of MS. There are positive effects of smoking cessation on disease progression, even after multiple sclerosis onset. It is imperative to prioritize interventions to reduce smoking rates in the general population. Moreover, targeting smoking cessation interventions specifically for individuals diagnosed with MS becomes crucial to impede the progression of the disease.

## 7. Alcohol

In a survey assessing health behaviors among 8983 participants in the NARCOMS Registry, 1636 (18.2%) participants were at risk for alcohol abuse or dependence. After age adjustment, 20.6% of participants were at risk for alcohol abuse or dependence [44]. Quesnel et al. showed that one in six MS patients drink excessively over their lifetime [71]. 

Several studies and meta-analyses showed no evidence that alcohol consumption is associated with an increased risk of developing MS, and some of them suggest a protective effect of alcohol consumption [72,73]. Hedström et al. investigated the possible association of alcohol consumption with the risk of developing MS in two large population-based case–control studies. Alcohol consumption exhibited a dose-dependent inverse association with MS in both sexes [74]. In a more recent study, the same authors analyzed data from the Swedish Epidemiological Investigation of Multiple Sclerosis. Their analysis included 2059 PwMS and 2887 matched controls. The authors found a dose-dependent inverse association between alcohol consumption and the risk of MS, with a 20% risk reduction for MS among alcohol consumers compared with never-consumers (OR 0.8; 95% CI 0.7–0.9) [75]. 

Similar results were found in a Danish case–control study on 1717 PwMs and 4685 healthy blood donors, indicating that alcohol consumption during adolescence was associated with a decreased risk of developing MS in both males and females [76].

In contrast, a study examining the premorbid lifestyle of 94 individuals with MS and 59 controls revealed that the MS group was more likely to have consumed alcoholic beverages occasionally or frequently and initiated alcohol consumption at an earlier age [77]. Moreover, significant associations between hard liquor and wine consumption were reported in a case–control study conducted in Belgrade [78].

Pakpoor et al. demonstrated an increased risk for MS following alcohol abuse, alcohol dependence, and alcohol use, with a significantly elevated risk for MS within 1 year of first admission for alcohol abuse [79]. 

Data on alcohol intake and MS severity and progression were somehow inconsistent. In a study by Ivashynka et al., ever-drinkers did not show a statistically significant association between alcohol intake and MS severity [80]. In a cross-sectional study conducted in Belgium on 1372 PwMS, a decreased risk of reaching EDSS 6 was found in regular consumers of alcohol in the relapsing-onset group. The majority of the participants in the study reported moderate drinking of alcoholic beverages and consumed 1–7 drinks weekly [81]. On the contrary, Pez-Ballesteros et al. demonstrated that ex-consumers of alcohol had a lower risk of progression than current consumers (HR 0.33; 95% CI 0.14–0.83) [82]. Quesnel et al. reported that individuals with MS who had a history of problem drinking had a higher lifetime prevalence of anxiety. Problem drinking was also significantly associated with a lifetime prevalence of suicidal thoughts, substance abuse, and a family history of mental illness [71].

Alcohol can have beneficial and detrimental effects on the pathogenesis of MS. Low and moderate alcohol consumption has been shown to attenuate innate inflammatory responses in humans [83]. It also induces apoptosis in oligodendrocytes and neurons, causing demyelination and affecting the CNS directly (Figure 2) [84].

Regarding the impact of alcohol on MS, research has yielded inconsistent findings. Some studies suggest moderate alcohol consumption may protect against developing MS or attenuating disease severity. However, other studies have not found a significant association between alcohol consumption and MS risk or disease progression. It is worth noting that the existing studies on alcohol and MS vary in design, sample sizes, methodologies, and measures used to assess alcohol consumption and disease outcomes. This heterogeneity makes it challenging to draw definitive conclusions from the available evidence.

## 8. Dyslipidemia

In population-based studies, dyslipidemia is one of the most frequently reported comorbidities in PwMS, with a prevalence of approximately 10% [22]. However, some studies reported a higher prevalence of dyslipidemia in MS patients. Edwards et al., in a large retrospective study, reported a prevalence of hyperlipidemia of 25.9% in PwMs, and they also found a higher prevalence of this comorbid condition in males compared with females [25]. In another large cohort of MS patients enrolled in the NARCOMS Registry, the prevalence of hypercholesterolemia was 37% [85]. In a cohort of 1124 male veterans with MS, hypercholesterolemia affected 49% of the subjects [29]. 

A study by Marrie et al. on 23,382 MS cases on the prevalence of comorbidity in this population at the time of MS diagnosis found that all comorbidities, except hyperlipidemia, were more common in the MS population compared to a matched population [28].

Multiple studies have explored the relationship between hyperlipidemia and disability progression or inflammatory activity in individuals with MS. Conway et al. conducted a longitudinal 3-year study on RRMS patients to examine the influence of hyperlipidemia, hypertension, diabetes, and obstructive lung disease on the disease course. Hyperlipidemia was the only factor that did not impact clinical outcomes, including walking speed, self-reported disability, and depression [33]. In contrast, Marrie et al. found that a diagnosis of hypercholesterolemia at any point during the disease course was associated with early gait disability and the need for ambulatory assistance [15]. The findings were unchanged in the sensitivity analysis restricted to RRMS patients [15]. Similarly, Tettey et al., in a prospective population-based cohort study, found that total cholesterol, apolipoprotein B, and the apolipoprotein B to apolipoprotein A–I ratio were independently associated with a higher EDSS score [50].

In a study by Weinstock-Guttman, higher high-density lipoprotein (HDL) cholesterol levels were associated with a lower probability of contrast-enhancing lesions and lower contrast-enhancing lesion volume at follow-up. Worsening of the EDSS score was associated with higher baseline low-density lipoprotein (LDL) cholesterol and total cholesterol levels [86]. However, this study had several limitations, including its retrospective design and the fact that the majority of patients were on different DMTs, and a small proportion of patients were treated with statins which could have acted as potential confounding factors [86]. 

In a study involving 18 consecutive patients with CIS who underwent monthly brain magnetic resonance imaging (MRI), blood sampling, and neurological assessment over a 6-month period, the authors discovered a significant correlation between the average number of enhancing lesions and the mean levels of total and LDL cholesterol [87]. Another study focusing on CIS patients treated with IFN-β revealed that higher levels of total and LDL cholesterol were linked to a greater cumulative number of new T2 lesions over a two-year period. This suggested a potential connection between an unfavorable lipid profile and increased inflammatory activity observed on MRI [88]. However, in a prospective study, lipid-related measures were not associated with the risk of MS relapse [51].

One possible explanation for these findings could be that chronic hypercholesterolemia might contribute to inflammation at the vascular endothelium level, leading to the induction of adhesion molecules and the recruitment of monocytes [89]. 

In summary, the results of studies examining the relationship between lipid profiles and MS show some inconsistencies, partly due to confounding factors resulting from different proportions of patients receiving treatment with cholesterol-lowering agents or the possible impact of DMTs on the lipid profile [90]. It is worth noting that statins, 3-hydroxy-3-methylglutaryl coenzyme A reductase inhibitors, are known to have immunomodulatory effects, including the inhibition of leukocyte migration across the BBB [91]. 

Several studies have been conducted to evaluate the use of statins in MS, yielding conflicting results but mostly suggesting a beneficial role [92]. In a single-arm study by Vollmer et al. involving 30 cases of RRMS patients, treatment with oral simvastatin at a dosage of 80 mg per day demonstrated a reduction of 44% in the number of gadolinium-enhancing lesions and a decrease of 41% in lesion volume compared to the pre-treatment period [93]. In a double-blind placebo-controlled trial involving patients with SPMS, oral high-dose simvastatin treatment reduced the annualized rate of whole-brain atrophy compared with placebo [94]. Nevertheless, conflicting findings have emerged, as some studies still need to provide evidence supporting the protective effect of statin therapy [95,96]. 

## 9. The Aging MS Population and Comorbidities: Implications

Life expectancy for PwMS has almost equaled that of the general population [97]. The aging process itself brings about various physiological changes that can predispose individuals to an increased risk of developing comorbid conditions [97]. In the case of MS patients, age-related comorbidities, especially of vascular origin, are becoming increasingly prevalent. Vascular comorbidities, such as hypertension, diabetes type-2, dyslipidemia, and cardiovascular disease, can significantly impact the clinical course of MS [15]. Furthermore, vascular comorbidities are of particular interest as they are mostly modifiable health states, along with voluntary behaviors such as smoking and alcohol consumption commonly observed among individuals with MS across all age groups [44]. Controlling and treating these cardiovascular risk factors might have a substantial impact on disease progression [44]. 

Vascular diseases have the potential to impact MS at a pathophysiological level. Alternatively, these vascular conditions might exert independent effects on disability, which add to the disability caused by MS. For instance, diabetic neuropathy can lead to gait impairment, with the combined effects of diabetes and MS leading to greater disability [15]. Vascular risk factors have also been implicated in the etiology of cerebral small vessel disease (CSVD). Typical MRI markers of CSVD in the brain include white matter hyperintensities, lacunes, microbleeds, enlarged perivascular spaces, and subcortical infarcts [98].

### Radiological Features of MS and Cerebral Small Vessel Disease (CVSD)

Vascular lesions, such as lacunar infarcts or white matter hyperintensities of presumed vascular origin, are commonly seen in patients of advanced age or with vascular risk factors. In the age group 45–55 years, white matter signal changes occur in more than 50% of asymptomatic individuals and can mimic MS lesions on MRI [99,100]. One of the unique challenges faced by the aging MS population is the difficulty in differentiating neuroimaging mixed patterns with vascular and MS lesions. Both conditions can exhibit comparable radiological features, complicating the differentiation process and treatment decision-making [99]. White matter hyperintensities of presumed vascular origin are hyperintense in the T2 and fluid attenuation inversion recovery (FLAIR) sequences; they typically occur bilaterally and symmetrically; their preferred localization is periventricular, located adjacent to the lateral ventricular wall or in the deep white matter [101]. The frontal and parietal lobes are the predilection sites of white matter lesions. In the supratentorial region, they are more common in the basal ganglia, corona radiata and centrum semiovale [98].

Lacunar infarcts are small-size infarcts (3 mm to 20 mm) caused by the occlusion of small, deep-penetrating branches of the large cerebral arteries [102]. Most lacunes occur in the basal ganglia, thalamus, subcortical white matter, and pons [102]. Lacunar infarcts are typically visualized as focal lesions characterized by increased T2-weighted signal intensity, and they can be clinically silent or, less frequently, symptomatic [103].

McDonald criteria revised in 2017 have proposed the spatial and temporal dissemination of MS lesions. Spatial dissemination requires white matter hyperintensity in at least two of the four typical sites: the periventricular, cortical, or juxtacortical, and infratentorial brain regions and the spinal cord. Dissemination in time requires the presence of gadolinium-enhancing and non-enhancing lesions in a single MRI scan or the discovery of new T2-hyperintense or gadolinium-enhancing lesions on follow-up MRI [104]. An area of hyperintensity on a T2-weighted MRI scan has to measure at least 3 mm in the long axis [104].

In general, conventional MRI techniques are not able to distinguish between demyelinating and ischemic lesions, but several radiological signs can help in the distinction. Dawson’s fingers are elongated, flame-shaped lesions perpendicular to the lateral ventricle wall, distributed along the axis of medullary veins. Dawson’s fingers are widely used as important imaging markers for the diagnosis of MS and to differentiate demyelinating MS lesions from non-specific white matter lesions and CSVD [105]. However, they are not specific to MS and can be found in patients with CVSD, particularly in those with diabetes [106,107].

Susceptibility-weighted sequences at 3 Tesla can identify paramagnetic rim lesions in around 50% of patients with MS. This feature, reflecting iron within phagocytes at the edge of chronic active lesions, rarely occurs in other neurological diseases and therefore has the potential to increase MRI specificity in differentiating MS from other disorders [108,109].

The central vein sign, which refers to the presence of a vein located centrally within brain white matter lesions, has been regarded as an imaging indicator for MS at standard clinical magnetic field strengths. Some studies show that the central vein sign can distinguish MS from mimic diseases, including CSVD [110,111]. However, the proportion of MS lesions showing the central vein sign and the most effective diagnostic threshold are higher when utilizing optimized susceptibility-weighted MRI sequences at high and ultra-high field strengths [112,113]. Despite these advancements, its routine clinical application is not yet recommended [108].

## 10. Conclusions

In this comprehensive review, we summarized currently available data on the prevalence of modifiable comorbidities and risk factors in PwMS and the possible effect of this association on MS course and disability progression. We also highlighted the unmet needs in this field.

In conclusion, recognizing and effectively managing cardiovascular risk factors in individuals with MS is of paramount importance for halting disease progression. By controlling hypertension, dyslipidemia, and obesity and promoting smoking cessation, healthcare providers can mitigate the impact of cardiovascular comorbidities on the neuroinflammatory processes, potentially improving clinical outcomes and enhancing the overall quality of life for PwMS. The demographic age shift in the MS population necessitates a heightened awareness of comorbidities, especially vascular disorders, that can significantly impact disease progression and management. Furthermore, differentiating between vascular and MS load lesions poses a significant challenge due to overlapping radiological features. Future research should focus on developing reliable diagnostic tools and strategies to accurately identify and manage comorbidities in the aging MS population, ultimately improving patient outcomes and quality of life.

## Figures and Tables

**Figure 1 jpm-13-01524-f001:**
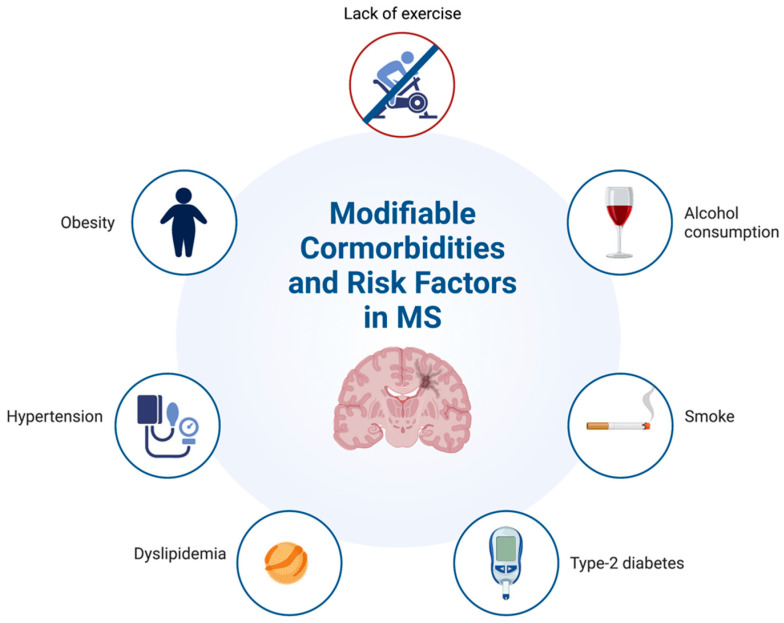
Modifiable comorbidities and risk factors in people with multiple sclerosis (MS).

**Figure 2 jpm-13-01524-f002:**
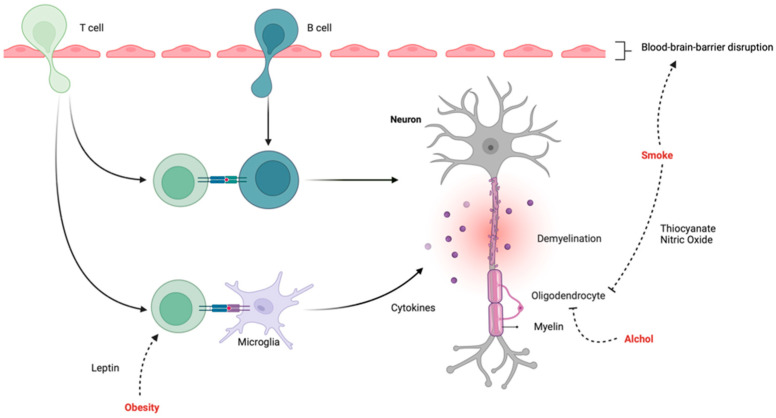
The complex interplay between modifiable risk factors and the pathogenesis of multiple sclerosis.

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
