# Peer review of "The Importance of Managing Modifiable Comorbidities in People with Multiple Sclerosis: A Narrative Review"

_jpm, 2023, doi:10.3390/jpm13111524_

Round 1

Reviewer 1 Report

Overall, the manuscript is well written and highlighting an important disease. I must say to add a diagram describing the mechanism of pathogenesis in multiple sclerosis and its comorbidity. So authors are requested to add mechanistic figure about the pathogenesis of Multiple sclerosis.

The English language quality is appropriate. 

Author Response

Reviewer #1 reports:

Overall, the manuscript is well written and highlighting an important disease. I must say to add a diagram describing the mechanism of pathogenesis in multiple sclerosis and its comorbidity. So authors are requested to add mechanistic figure about the pathogenesis of Multiple sclerosis.

We thank the referee for his/her comment, and accordingly, we added a Figure on MS pathogenesis and the interplay with comorbidities.

Reviewer 2 Report

This review describes the current evidence on the range of preventable and modifiable comorbidities and risk factors and their implications for people with MS.

In principle, the topic is very interesting, and the idea of developing an article is good.

The problem is that the methodology used is not valid.

If you want to do a review, it must be done with an appropriate methodology.

A systematic review would be appropriate.

The title should say that it is a review.

In the summary the objective of the research.

In the methodology, the criteria for inclusion of articles, and its prism method.

In the results, they are not grouped by pathologies.

in the conclusions, what is extracted from the review, and that contributes to science.

We recommend doing a systematic, scientific review.

Author Response

Reviewer #2 reports: This review describes the current evidence on the range of preventable and modifiable comorbidities and risk factors and their implications for people with MS.

In principle, the topic is very interesting, and the idea of developing an article is good.

The problem is that the methodology used is not valid.

If you want to do a review, it must be done with an appropriate methodology.

A systematic review would be appropriate.

The title should say that it is a review.

In the summary the objective of the research.

In the methodology, the criteria for inclusion of articles, and its prism method.

In the results, they are not grouped by pathologies.

in the conclusions, what is extracted from the review, and that contributes to science.

We recommend doing a systematic, scientific review.

Thank you for taking the time to provide your feedback on our article. We appreciate your interest in the topic and your valuable input. We acknowledge your concerns regarding the methodology used in our review. We want to clarify that the purpose of our review was not to conduct a systematic review but rather to provide a descriptive overview of the current evidence on preventable and modifiable comorbidities and risk factors in people with MS. We intended to offer a comprehensive understanding of the topic without adhering to the strict criteria of a systematic review We understand the importance of systematic reviews in scientific research, and we appreciate your suggestion to conduct one in the future. However, for this article, we aimed to present a broad overview and synthesis of existing evidence without the limitations imposed by a systematic review methodology.

As recommended by the reviewer we stated in the title that the review is a narrative one.

Reviewer 3 Report

The Manuscript "The importance of managing modifiable comorbidities in people with multiple sclerosis" is really interesting and well written. 

All the information provided are relevant and bring new insights on MS risk factor. 

Introduction: 

The introduction provide an in-depth review on the disease and discuss the importance of the risk factors on the onset of the disease. 

I would suggest to develop the section on the immune system. The latter plays a major role in the course of the disease. It has a detrimental role but also microglia helps to clear myelin debris and allows "re myelination". I would be also interesting to mention the current treatments and how they can modify the disease course. 10.3389/fncel.2020.00284 I found it helpful. 

In the body of the manuscript, I would suggest also to briefly discuss the effect of comorbidity / risk factor on the immune system. Obesity, alcohol, smoke, ... trigger an inflammatory response, also If the liver is affected, the immune system can be stimulated. 

Overall, the manuscript is really interesting and provide new insights on the disease. These comments are only minors

Thanks

Author Response

Reviewer #3 reports:

The Manuscript "The importance of managing modifiable comorbidities in people with multiple sclerosis" is really interesting and well written.

All the information provided are relevant and bring new insights on MS risk factor.

Introduction:

The introduction provide an in-depth review on the disease and discuss the importance of the risk factors on the onset of the disease.

I would suggest to develop the section on the immune system. The latter plays a major role in the course of the disease. It has a detrimental role but also microglia helps to clear myelin debris and allows "re myelination". I would be also interesting to mention the current treatments and how they can modify the disease course. 10.3389/fncel.2020.00284 I found it helpful.

In the body of the manuscript, I would suggest also to briefly discuss the effect of comorbidity / risk factor on the immune system. Obesity, alcohol, smoke, ... trigger an inflammatory response, also If the liver is affected, the immune system can be stimulated.

Overall, the manuscript is really interesting and provide new insights on the disease. These comments are only minors

We thank the referee for his/her valuable comments.

Accordingly, we added in the text a sentence on the beneficial role of neuroinflammation in MS: “It is plausible that the neuroinflammatory component remains a major driver of MS pathogenesis, causing myelin sheath destruction and, ultimately, axonal damage. However, a significant number of evidence has shown that inflammation is not entirely detrimental. Previous studies have documented the reparative activities of the inflammatory response in MS. In these circumstances, immune cells release neurotrophic factors and engage with neurons and glial cells to protect them from damage, promoting growth and repair”.

Furthermore, we added, as suggested, a sentence on the current treatment: “The treatment scenario of MS has changed significantly in the past decades due to the growing number of approved disease-modifying therapies (DMTs), which can influence the disease course by preventing relapses and disability progression in people with MS (PwMS). Furthermore, with a better understanding of the pathophysiological path-ways that drive progressive disease, immunomodulatory therapies that simultaneously reduce acute and chronic neuroinflammation are actively being studied in later-phase clinical trials for both relapsing and progressive MS. And several other strategies have been studied to target remyelination”.

Furthermore, we added a Figure to explain the interplay between comorbidities and the immune system in MS.